# Entry Mode, Market Selection, and Innovation Performance

**Chong Wu** [1,2], **Fang Huang** [1,*], **Caihong Huang** [1] and **Huiming Zhang** [1]

1   Business School, Nanjing University of Information Science & Technology, Nanjing 210044, China;
    002063@nuist.edu.cn (C.W.); 20161263737@nuist.edu.cn (C.H.); 002068@nuist.edu.cn (H.Z.)
2   Business School, Nanjing University, Nanjing 210023, China
*   Correspondence: huangfang@nuist.edu.cn

**Abstract:** Recent studies highlighted the need for multi-perspective research on the internationalization and performance of emerging market multinational enterprises (EM-MNEs) and revealed why many EM-MNEs perform negatively when they respond to the host-country environment. Using a sample of listed Chinese manufacturing firms from 2003 to 2014, this study examines the relationship between entry mode choice, which is driven by different environmental response patterns, and firms' innovation performance. We further analyze the moderating role of market selection on the relationship between host-country institutional factors and firms' innovation performance. The results show that the international breadth of firms and the economic stability and investment protection of the host country significantly promote firms' innovation performance. While the entry mode is unilaterally driven by the host-country response, early international experience and the international depth of firms have significant negative effects on firms' innovation performance. The level of economic development in the invested area plays a moderating role in the relationship between the host-country institutional factors and firms' innovation performance. Our findings enrich the literature on the relationship between internationalization and EM-MNE performance, and provide inspiring and straightforward empirical evidence.

**Keywords:** EM-MNEs; entry mode; market selection; institutional environments; parent innovation performance

## 1. Introduction

With the rapid development of emerging market multinational enterprises (EM-MNEs) outward foreign direct investment (OFDI), the Chinese manufacturing industry shifted from the internal internationalization stage, which features the introduction of technology, to the external internationalization stage driven by OFDI. This is a critical stage of "external" transformation and involves an upgrading strategy. In recent years, OFDI in China's manufacturing industry accelerated, showing the characteristics of international leapfrog development. During the 12th five-year plan period, for example, the OFDI plan (which includes the financial sector) involved $500 billion, while the actual amount spent was $521 billion, exceeding expectations by 4.2%. In the first year of the 13th five-year plan, OFDI exceeded $170 billion, which is an increase of 44.1% from the year before, and the OFDI of the manufacturing industry is up to 18.3%. This change shows that, in the absence of a gradual accumulation of internationalization through "commodity exportation, sales subsidiary, investment and setting up factory", many manufacturing firms carry out a large amount of OFDI due to the urgent need for internationalization and the encouragement of local governments. In terms of entry mode, using a joint venture in mergers and acquisitions (M&A) has become the main mode of OFDI. In terms of market selection, investment in the past mainly focused on the invested location

in terms of the physical supply of energy and mineral products, but it now focuses on developed economies that have strategic resources such as technology and brands. However, in contrast to the size and speed of OFDI, the internationalization capability of firms generally lags behind the growth in overseas markets. According to the statistics published by the China Enterprise Confederation and the Chinese Entrepreneur Association, the average transnational index of China's top 100 MNEs in 2017 was 14.85%, which is 0.45% higher than the previous year. However, the average transnational index of 14.85% is not only far lower than the average transnational index of 64.55% for the world's top 100 MNEs, but also far lower than the average transnational index of 54.22% for the top 100 MNEs of developing countries.

Based on the data described above, there are three characteristics and problems of China's manufacturing enterprise OFDI that need attention. Firstly, due to the urgent need for internationalization and the fact that local governments encourage internationalization, many manufacturing firms carried out a large amount of OFDI without the gradual accumulation of the internationalization of "commodity exportation, sales subsidiary, investment and factory setting up". Secondly, the two types of standard strategic choices in terms of the entry mode/market selection used for OFDI also indicate that there are two typical characteristics of latecomers in emerging economies: (1) in terms of the choice of entry mode, the structure and practice of decision-making regarding the entry mode reflects the trend of "institutional isomorphism" in the host country, and when responding to host-country "legitimacy", increasing numbers of firms use a joint venture with local firms in M&A. (2) In terms of market selection, the investment region reflects the tendency of reverse investment in response to the "innovation strategy". Increasing numbers of firms are shifting OFDI from Asia and Africa to developed economies such as Europe and the USA to effectively achieve global synchronization in technological innovation. Finally, with the acceleration of the OFDI process, the development of firms' internationalization capabilities lags behind the demand for technological innovation and even causes existing behavior patterns and organizational practices to be misaligned with the current needs related to research and development (R&D) internationalization, which restricts the effectiveness of their innovation strategy implementation.

Obviously, Chinese MNEs have comparative advantages, such as a growth in OFDI in China's economic transformation process, unique institutional environments, and the leap-forward development of firms. China provides a good context to study the development of EM-MNE internationalization and performance, and this study may enhance theoretical integration and development [1]. Based on the paper by Keith Brouthers' [2], which won the Journal of International Business Studies (JIBS) 2012 Decade Award, Shaver [3] discussed the research on entry mode and the subsequent performance, and further proposed suggestions regarding model explanatory power, data measurement, etc. He did this because, with the development of globalization and intensified international competition, MNEs changed their traditional strategic choices. Seeking economic efficiency driven by the transaction cost (TC) perspective will not help organizations gain sustained competitive advantages in boundary expansion, which proved the limitations of TC theory [4]. The resource capability (RC) perspective and the institutional-based (IB) perspective supplemented TC theory in terms of firm internationalization. From the RC perspective, MNEs cannot just be confined to their existing capability. Instead, their capabilities should be developed through the innovation of entry mode. Consequently, the RC perspective frees the constraints of opportunism in TC theory, which provides the corresponding theoretical support for the strategic choice of inter-firm cooperation or joint ventures [5]. Meanwhile, with the development of economic globalization and information technology, the complex uncertainty caused by MNEs' multiple institutional environments increased the effects of their strategic choices and the organization mechanism. Increasingly, scholars from the IB perspective criticized the old TC paradigm that neglects the moderating roles of market selection and the host-country institutional environment on the relationship between EM-MNEs' strategic choices and their performance. In this respect, Hennart and Slangen [6] particularly emphasized the significant weakness of the operational mode of EM-MNEs in responding to host-country legitimacy. It can be

seen that the scholars in this field are concerned about the influence of international capabilities and the response to the host country on EM-MNEs' innovation performance in terms of entry mode/market selection.

In the eclectic paradigm, market selection and choice of entry mode are treated as one decision, and the emphasis is on choice of entry mode. The need for information about the market is the crucial indicator for the choice of entry mode [7]. Overall, some scholars believe that it is only possible to reveal the relationship between the strategic choices and the subsequent performance of MNEs by incorporating entry mode and market selection into an integrated analytical framework [8]. This is because, once MNEs enter the overseas market, the entry mode choice is driven by the organization's response to internal consistency and the external environment, which influence innovation performance. The relationship between organizational behavior and performance is moderated by market selection; in other words, the socioeconomic conditions of the invested area moderate the relationship between host-country institutional factors and MNE innovation performance [9]. Therefore, in the leapfrog development stage, the principles related to the direct impact of entry mode and the moderating effects of market choice on innovation performance should be synthetically tested. Suggestions can be made to help MNEs simultaneously develop internationalization capabilities and make use of the external environment.

The contributions of this paper are as follows: firstly, this study matches the firms that had OFDI projects from 2003 to 2014 in the "Overseas Investment Firm list" published by the Chinese Ministry of Commerce with "industrial firms in the database of Shanghai and Shenzhen A shares" in the same period to select samples of listed firms in the manufacturing industry with OFDI experience. Although a very limited number of samples are available, this study can provide important data and propose methods that can be used for follow-up research on EM-MNEs.

Secondly, from the perspective of dual responses to internal consistency and host-country consistency, this paper combines entry mode/market selection and innovation performance in a theoretical framework. This study analyzes the impacts of the entry mode using a prediction model for firms' innovation performance, and analyzes the moderating roles of market selection, i.e., the level of economic development on the relationship between institutional factors and firms' innovation performance.

Finally, the internationalization factors of Chinese firms that impact innovation performance are measured, and the results reveal the different influences of internationalization factors on innovation performance. This study examines straightforward empirical evidence and suggests future research ideas for research on the relationship between MNEs' internationalization and their performance.

## 2. Literature Review and Hypothesis Development

### 2.1. Group Response and Entry Mode Choice

The TC perspective holds that the TC caused by an incomplete market is a main influencing factor for MNEs' entry mode choice. It is necessary for MNEs to make the optimal choice from among the organizational structures' market governance, hierarchical governance, and hybrid governance to maximize their economic efficiency [10]. When the transaction is complicated, the TC perspective suggests that MNEs adopt the wholly owned subsidiary (WOS) entry mode; in this mode, MNEs pay fewer transaction costs to be involved in resource allocation activities [2]. The RC perspective indicates that the tacit and sticky nature of MNEs' specialized knowledge can lead to two trading parties symmetrically absorbing and integrating knowledge. The more specialized the MNEs' knowledge and technology assets are, the more likely it is that MNEs will select the WOS entry mode [11]. The IB perspective also claims that asset specificity is an important internal institutional factor that influences TCs and determines the choice of MNEs to incorporate either corporate governance or market governance. An increase in the degree of asset specificity will result in the value loss of fixed assets or technical assets that were created for alternative uses, thereby increasing the opportunistic

risk. In summary, when the level of asset specificity is high, MNEs will use higher-control entry modes, such as the WOS mode, to improve efficiency or avoid risk [4].

From the perspective of internalization and real options, it is believed that the strategic flexibility of MNEs has a significant effect on the risk avoidance of and overall value increase in MNCs in the international market. This occurs because MNEs deeply explore the market of a country or of a few countries, and they operate more widely in multinational markets, forming an institutional inertia of intra-group resource allocation and integration. The institutional advantage of such an internal market can lead to the flexible deployment and integration of resources in accordance with changes in the international market [12]. Thus, to make use of the advantages in terms of the marketing strategic flexibility of these countries, MNEs tend to use the WOS mode to deploy resource groups to reduce risk and increase their overall value [13].

Cherity and Eriksson [14] argue that early internationalization increases the empirical knowledge of MNEs in foreign markets. Erikkson et al. [15] classified empirical knowledge obtained from foreign markets into three dimensions: international knowledge, foreign business knowledge, and foreign institutional knowledge. Foreign business knowledge refers to empirical knowledge of customers, competitors, and the market. MNEs also need empirical knowledge regarding their resources and capability of conducting operations in foreign markets. International knowledge can support MNEs operating in a variety of situations and will help them continually expand their knowledge and skills. Moreover, in the process of MNEs' acquiring market knowledge by investing in resources in foreign markets, international market knowledge and experience and foreign institutional knowledge accumulate, which affect MNEs' decisions regarding how to invest further in foreign markets [16]. International business scholars generally believe that MNEs' international experience can also influence international decision-making, and the accumulation of experience will increase the possibility of joint ventures (JVs) or alliances [17,18].

The innovation orientation of MNEs can also affect their entry mode choice. The innovation orientation forms organizational inertia that determines how MNEs will use and integrate their resources and capability [19]. Under competitive pressure, MNEs' strategies would shift to "capability development". Considering the limitations of their ability to acquire and integrate knowledge in the process of capability development, MNEs tend to use JV or alliance modes for risk-sharing and to obtain complementary advantages. The fast-growing "inter-organization" or "strategic alliance" cannot be simply seen as a low-cost or efficient alternative to other entry modes, but rather as an important method that can be used for MNEs' knowledge acquisition, knowledge integration, and capability development [5]. Empirical analyses show that MNEs prefer the WOS mode in the process of capability utilization, but tend to choose the JV entry mode in the process of capability development [20].

Finally, the institutional and resource integration view indicates that ownership is the basis of Chinese firm-level institutions. On the one hand, although some privately owned firms form a modern system through public listing, they still use a family style of management and institutional tendencies in nature, and are inclined to use the highly controlled internally integrated development mode; thus, they have a strong tendency to choose the WOS entry mode [21]. On the other hand, China's state-owned firms have more political goals and tasks, and the government also has constraints in terms of the allocation of power in their corporate governance structure; for example, major investment and decision matters are overseen by the Chinese government and must be determined by the leading group. The decision-making and executive incentives are also constrained by the abovementioned political objectives and tasks. Thus, state-owned firms are frequently tasked with the acquisition of strategic resources, and they are generally inclined to adopt the JV mode because it can be accomplished in a short amount of time, and this mode is highly efficient for meeting the abovementioned objectives and tasks as soon as possible [21,22]. Thus, we propose the following hypotheses:

**Hypothesis 1 (H1).** *The more pressure there is for responding to group consistency, the more likely an EM-MNE will be to choose the WOS entry mode.*

**Hypothesis 1.1 (H1.1).** *Compared with the JV entry mode, the greater the asset specificity is, the greater the strategic flexibility, the greater the tendency of the firm to be a private firm, and the greater the likelihood that the EM-MNE will use the WOS mode to enter the foreign market.*

**Hypothesis 1.2 (H1.2).** *Compared with the WOS mode, the greater the international experience is, the greater the innovation orientation, and the greater the likelihood that the EM-MNE will use the JV mode to enter the foreign market.*

*2.2. Host-Country Response and Entry Mode Choice*

Some economists emphasize the importance of technology, capital, and labor for economic growth; however, according to institutional economists, this occurs because of institutional action. Moreover, institutional economists believe that institutions are not only the root cause of economic growth, but also the root cause of economic fluctuations. Although other factors cause economic fluctuations, the institution is the most fundamental reason [23]. The institution includes price mechanisms, supply and demand mechanisms, and competition and risk mechanisms. Existing studies generally claim that the more perfect economic foundation-related institutions are, the higher quality a country's institutional environment will be and the more stable the economic operation will be. Thus, international business scholars believe that MNEs can flexibly choose between the WOS mode and the JV mode according to the trends regarding the host country's economic uncertainty to achieve market growth or capability development. Particularly, when economic uncertainty regarding prices, exchange rates, supply and demand, etc. is small, MNEs usually choose to engage in market expansion or increase the efficiency of new technology development through the WOS entry mode. When economic uncertainty is high, MNEs prefer the JV mode, which can benefit their learning and capability development [24,25].

In addition, according to the IB perspective, Brouthers et al. [4] supplemented the entry mode theory with the TC perspective and explained how MNEs' transaction costs can impact their entry mode choice when it is based on the host country's investment uncertainty. The degree of legal perfection in the host country can affect the TCs of MNEs operating in foreign markets and, thus, have a significant impact on MNEs' choice of entry mode [26]. The more perfect the legal system of the host country is (i.e., the protection of property rights), the lower the TC of MNEs adopting the market mechanism is. This encourages MNEs to adopt entry modes that respond to internal consistency, such as full acquisitions or the WOS mode [26]. In contrast, the imperfect legal system of the host country can produce an ineffective market mechanism and higher TCs. To avoid risk, MNEs tend to replace the market mechanism with higher TCs with the JV mode. Moreover, countries with poor legal systems may also limit their local investment in MNEs, for example, by limiting the capability of MNEs to become monopolies or by helping local firms enhance their competitiveness through JV [27].

A firm's entry mode choice needs to not only satisfy its own interests, but also take into account the behavioral patterns and values in host countries [28]. Because formal institutions and informal institutions differ, scholars empirically examined the influence of institutional distance on of MNEs' entry mode choice. These scholars divided institutional distance into three types: regulatory distance, normative distance, and cognitive distance. Regulatory distance relates to formal institutions and reflects differences in the legal environment of countries; normative distance reflects differences in the social norms of countries; and cognitive distance reflects the sharing of beliefs between countries and the self-evident differences between mindsets [29]. In practical research, scholars define the distance between social norms and cognition as normative distance, which belongs to the informal institution category, and they use cultural distance to quantify the informal institution distance between home and host countries. A sample study of transition economies conducted by Estrin et al. [30] shows that the greater the cultural difference is, the more likely MNEs are to choose the JV mode over the WOS mode. Brouthers [2] also found that the greater the cultural distance is, the greater the uncertainty, the higher

the knowledge barriers, and the higher the management and operating costs of MNEs. Thus, MNEs are more willing to use the JV mode to enter local markets. Thus, we propose the following hypotheses:

**Hypothesis 2 (H2).** *As pressure for responding to the host country's institution increases, EM-MNEs will be more inclined to choose the JV entry mode.*

**Hypothesis 2.1 (H2.1).** *The greater the host country's economic uncertainty, the greater the cultural difference between the home and host countries, and the greater the likelihood that EM-MNEs will use the JV mode to enter the foreign market rather than the WOS mode.*

**Hypothesis 2.2 (H2.2).** *The more perfect the host country's legal system is, the greater the likelihood that EM-MNEs will use the WOS mode to enter the foreign market rather than the JV entry mode.*

*2.3. Entry Mode Choice and Firm's Innovation Performance*

Entry mode theory assumes that a reasonable entry mode choice enables a firm to improve its performance [4,31]. It is suggested that the entry mode choice driven by the TC perspective can provide a firm with the most effective structure [32,33]. Because TC theory indicates that a firm will choose the most effective structure from a set of substitutable modes to achieve the best performance, the firm may underperform and eventually be driven out from a competitive industry if it fails to consider TC theory and chooses a less effective entry mode [27,34]. However, some scholars claim that the entry mode choice driven by the TC perspective may not lead to the best organizational performance, and the inappropriate use of TC theory can also have a negative influence on firm performance [5,35]. On the one hand, the entry mode choice driven by TC perspective focuses too much on cost minimization and ignores the firm's value creation [36,37]. The RC perspective holds the view that rapidly developing "inter-firm organizations" or "strategic alliances" cannot be simply deemed a low-cost or highly efficient substitute for other entry modes, and these modes could be important methods for MNEs to engage in knowledge integration and capability development [38]. On the other hand, the entry modes driven by the TC perspective often underestimate the location factors, especially the institutional factors of the host country [2]. This occurs because, in the institutional framework, the foreign subsidiaries of MNEs not only exist in the institutional environment of the host country, but also in the firm-level institutional environment [39–41]. This type of dual institutional environment often creates conflicts, posing many difficulties and challenges for MNEs. As such, research on how MNEs simultaneously respond to the conflicting pressures of the dual institutional environment needs to be developed in the future [42–44].

Research on the entry modes and performance from the IB perspective still seem vulnerable in terms of situational adaptability. Many scholars use the same theoretical elements and test variables in different contexts, which leads to conflicts in terms of the relevant conclusions of the empirical tests. Thus, some scholars suggest that research on MNEs' entry modes and performance should be divided into two aspects, namely "MNEs in developed countries entering emerging markets" and "EM-MNEs entering the markets of other countries" [45]. In the former, MNEs in developed countries have some monopolistic advantages, and their strategic target is focused on market expansion and knowledge transfer. Previous research on the TC entry mode provides some support for this situation; for example, Brouthers et al. [4] found that firms using entry modes predicted by TC theory performed statistically significantly better than firms using other "misaligned" modes. Nevertheless, in the latter, EM-MNEs enter foreign markets through their subsidiaries, and they have a strong will to integrate into the global innovation network and develop their innovation performance through subsidiaries. EM-MNEs enter the foreign market by making reasonable choices in terms of entry mode; their subsidiaries make use of local institutional environments to integrate local resources for technological innovation, and the subsidiaries can continuously develop their innovation ability to become valuable contributors to group innovation. This process occurs because these subsidiaries are overseas "listening posts" of the

EM-MNEs; when they capture advanced knowledge and technology, they will facilitate EM-MNEs' innovation activities by transferring knowledge to the parent [46]. Empirical research also showed that EM-MNEs are actively entering the global innovation network of knowledge and intangible resources through their subsidiaries, and this process continues to enhance EM-MNEs' knowledge accumulation ability and competitiveness [47,48]. In this type of situation, the previous scholarly literature established that the response of non-TC factors (such as the economic system and rule of law) to the host country's institutional environment are likely to have important consequences on the MNEs' mode choice and performance [4].

Therefore, when scholars study entry modes and performance based on the goal of enhancing EM-MNEs' innovation capability, they found that the influence of TC theory, which pays attention to internal advantages, is weakened. In contrast, both the RC perspective that is based on MNEs' innovation and development and the IB perspective based on the moderating role of the institutional environment become particularly important [49]. Meanwhile, the influence of the host country's economic and institutional environment on innovation performance should be highlighted [45]. Current research should also consider the comparative advantages of the above perspectives and use the IB perspective to integrate the theoretical elements of the TC and RC perspectives to focus on EM-MNEs' entry mode choices and innovation performance [25]. Brouthers et al. [4] already found a theoretical model that integrates the factors of responding to group consistency from the RC perspective and the factors of responding to host-country consistency from the IB perspective that can better predict MNEs' entry mode choices. An entry mode that is in line with the prediction of the above integrated theoretical model already increased the financial and nonfinancial performance of MNEs [31]. Based on this discussion, we hypothesize the following:

**Hypothesis 3 (H3).** *The entry mode that responds to group consistency and host-country consistency has a positive influence on firms' innovation performance.*

### 2.4. Market Selection and Firms' Innovation Performance

Recent studies showed that firms in developing countries are actively expanding overseas to select a better market environment in hopes of significantly enhancing their innovation performance and global competitiveness [50]. A well-developed market environment can provide firms with innovation resources that firms can rarely obtain on their own, enabling firms to gain support for innovative projects and institutional platforms (risk investment, strategic alliances, and R&D) [51]. For example, EM-MNEs are reversely investing in the markets of developed countries and can help them both avoid the technological block and obtain good innovation resources and advanced technologies. Developed countries have abundant innovation resources, well-developed intellectual property law, stable economic and market environments, and well-developed investment protection systems. It is reasonable to conclude that EM-MNEs in developed countries have significantly better conditions for competition and innovation than those in developing countries [52].

EM-MNEs tend to establish subsidiaries in markets that have a favorable institutional environment for acquiring knowledge and technology [53]. These subsidiaries can help EM-MNEs obtain strategic resources and specific competitive advantages from developed countries and increase their capability for technologically integrative innovation [54–56]. When they are in economically and institutionally well-developed countries, EM-MNEs' subsidiaries can recruit or cooperate with experienced talent, such as scientists, designers, and engineers, which can considerably enhance parent innovation performance [57,58]. Evolutionary theory regarding MNEs argues that acquiring and assimilating advanced knowledge and technology through OFDI can significantly increase knowledge, ultimately contributing to enhancing EM-MNEs' innovation performance [11]. In addition, a host country's good economic and institutional environment that facilitates free competition can promote dynamic inter-firm competition in the market. A good institutional environment can enable EM-MNEs' subsidiaries to continuously update their operation and innovation capabilities and help EM-MNEs

transfer innovation knowledge and technology to the parent while guaranteeing their own survival and development [59,60].

Existing studies generally claim that the more developed a host country's economic institution is, the better its investment protection and the more stable its economy. Therefore, a developed economic system and institutional environment can smooth the risks while firms follow their innovation strategies, which have long investment cycles. A stable economic environment and good investment protection will enable EM-MNEs to have better parent innovation performance [27,52], because a stable economic environment and well-developed investment protection enable the EM-MNEs' subsidiaries to have a better chance of integrating innovation resources, reducing risks in the process of innovation, and effectively transferring knowledge and capabilities within the group networks of EM-MNEs. However, compared to EM-MNEs that invest in developed countries, those that invest in developing countries are more sensitive to the roles of the host country's economic stability and investment protection during the innovation process [27]. In contrast to developed countries, the economy and investment protection in developing countries are relatively unstable; additionally, developing countries have less economic and institutional support for development, and they have a relatively large gap between economic and institutional development. Therefore, the instability of the economy and imbalance in investment protection may significantly increase EM-MNEs' investment risks and affect their innovation performance [61]. Thus, improvement in developing countries' economic stability and investment protection can have a visible positive influence on the innovation process and knowledge transfer in the networks of EM-MNEs [52]. It can be concluded that, compared with investment in developed countries, the socioeconomic conditions of developing countries can play a more positive moderating role in the relationships among economic uncertainty, investment protection, and parent innovation performance. Therefore, we propose the following hypotheses:

**Hypothesis 4 (H4).** *Improvement in economic uncertainty and investment protection in the host-country market has a positive impact on firms' innovation performance.*

**Hypothesis 5 (H5).** *The socioeconomic conditions of developing countries can play a more positive moderating role in the relationships among economic uncertainty, investment protection, and EM-MNEs' innovation performance than the markets in developed countries.*

## 3. Methodology

### 3.1. Sample

Our sample comprises Chinese firms publicly listed on the Shanghai and Shenzhen stock exchanges during the period from 2003 to 2014 (inclusive). The data collection process included three steps. Firstly, we collected foreign entry events of Chinese OFDI from the Ministry of Commerce of China (MOC). The MOC is a major ministry at the central government level that promotes and manages Chinese OFDI [50]. Every new OFDI project conducted by Chinese firms needs to be registered with the MOC. This data source provides a brief profile of each OFDI project (e.g., investment location, industry, date of approval, and line of business). We selected Chinese firms from this database. Secondly, we matched the foreign entry events of Chinese publicly listed firms to the "manufacturing firms" (code industry) from the China Stock Market and Accounting Research (CSMAR) databases, which were compiled by Shenzhen GTA Education Tech Ltd., a leading financial database provider in China. Finally, we excluded the following entries from our final sample: (1) manufacturing firms that had no OFDI experience; (2) samples that failed to define their business scope or completely disclose their R&D, sales, and intellectual property data in their annual reports for three years after overseas investment; and (3) listed firms that experienced asset restructuring and a change in their main business three years after overseas investment. As a result, we selected 242 samples from 178 listed manufacturing firms that had relatively complete data. At the same time, taking into account the predisposition of the influence of some variables such as early international experience, we used the value of such

explanatory variables the year before OFDI. Taking into account the lag of firms' R&D investment and innovation performance, the value of such variables was measured three years after OFDI. This study's data index used the span of 2002–2016 because of these lags and differences in the years that the variables were measured.

Of our samples (the entry area included Hong Kong, Macao, and Taiwan) that conducted OFDI from 2003 to 2014, 95 were state-owned firms and 147 were private firms, while 143 had invested in developed countries and regions, and 99 had invested in developing countries and regions. Most of the firm-level data and indicators were manually collected from the annual reports of the listed firms. Valid patent data were mainly collected from the annual reports of the listed firms and the China patent full-text database of China's State Intellectual Property Office, which provides data such as information on the patent applications, the number of patents granted, and the patent assignee. The two sources mentioned above can supplement each other. For macro-level data on the host countries, the indicator for the institutional level of investment protection was collected from the International County Risk Guide (ICRG) published by the Political Risk Services (PRS) Group. The indicator for economic uncertainty was measured using the rolling variance of the annual growth rate of the host countries' gross domestic product (GDP) for a seven-year period; these data were collected from the World Bank. Cultural distance refers to cultural differences, and we used Hofstede's five dimensions of culture to measure the cultural distance between the home country and the host country.

To examine the relationship between entry mode and EM-MNEs' performance, following Shaver [62] and Brouthers et al. [31], we included a control variable to correct for self-selection using Heckman's procedure in a performance regression equation. This variable, self-selection correction (also known as the "inverse Mills ratio"), was included because other studies provided evidence that unobservable firm characteristics can affect both mode choice and performance [4].

### 3.2. Dependent Variables

Two different explained variables were used in the research. Firstly, the market entry mode variable was used because this study focuses on the OFDI of Chinese firms. Instead of taking traditional market entry modes such as export and franchise into consideration, this paper adopted dummy variables for WOS and JV to indicate when these two market entry modes were used for the OFDI of Chinese manufacturers; 1 denotes the WOS mode, while 0 denotes the JV mode. This paper also focuses on the effect of internationalization on the innovation performance of Chinese MNEs, and we used the natural logarithm of the number of annual patent applications of Chinese MNEs' parents to indicate innovation performance. This measure can comprehensively and effectively reflect the quality of parent's innovation results. Additionally, to avoid losing observations with no patents, we added one to the actual values when calculating the natural logarithm [63,64].

### 3.3. Independent Variables

For the variables measuring the group response, the owner was set as a dummy variable: private firms are denoted by 0, while state-owned firms are denoted by 1. Asset specificity was divided into materials, places, brands, and human resources, and the level of investment in durable assets is a general indicator that is used to assess asset specificity [65]. Therefore, this paper used the ratio of fixed assets to total assets at the end of the year before OFDI. Considering MNEs' strategic choices and path dependence on organizational behavior, it is noteworthy that MNEs' international experience has a great influence on entry mode choice and performance [5]. In alignment with the general approach used by international business research, early international experience was measured by the ratio of foreign sale revenue to total sale revenue (i.e., FSTS) in the year prior to OFDI. Nevertheless, the accumulation of international experience is a step-by-step process. The Uppsala model constructed by scholars from the Nordic school is a typical example of methods used to calculate this variable. A firm first sets up an export business; then, it establishes subsidiaries to sell products in the overseas market, and finally, it enhances its level of investment by setting up plants in the overseas market. In this way,

the internationalization level of the firm moves from the period of export trade to OFDI, and the firm's capability is also enhanced because it applies domestic experience to running a business in the overseas market. In past research, the accumulation of internationalization experience was often measured by FSTS, but this indicator is an overall indicator of internationalization experience, including the overall experience of export trade and OFDI. Therefore, to further examine the relationship between internationalization and performance, the amount of international experience that the firm had after it engaged in OFDI needs to be measured. In addition, because a single indicator cannot accurately capture all the features of international experience, the degree of international experience can be further divided into the breadth of international experience, which reflects the degree of multi-regional investment degree, and the depth of international experience, which reflects the degree of repeat investment in specific areas, which can be measured by the number of foreign countries in which MNEs develop their first subsidiaries and the number of times that MNEs set up a subsidiary in the same country, respectively [12].

Meanwhile, we used the ratio of the breadth and depth of international experience to measure the influence of MNEs' strategic flexibility on their choice of entry modes. The ratio suggests that, when international experience is broader, the flexibility of resource integration through multi-markets is higher; when international experience is deeper, the flexibility of resource integration through multi-markets is lower. We measured MNEs' innovation orientation with the ratio of the number of R&D personnel to the number of all staff three years after OFDI, because measuring the increase in the number of research personnel after OFDI is a reasonable and effective way of calculating a firm's innovation orientation [19]. In addition, R&D input is measured by the ratio of R&D expenditures to operation income of MNEs three years after OFDI.

In terms of variables that respond differently to the host country, existing studies generally claim that when the economic system in a host country is more perfect, that system will be more stable, and the economic fluctuations are smaller; otherwise, there will be a large fluctuation in demand and supply. We used GDP-related data obtained from the World Bank and the rolling variance of GDP growth rate over a seven-year period to measure economic uncertainty, because short economic fluctuation cycles in different countries are normally shorter than seven years. The development of law in the host country is measured by "rule of law" in the World Governance Indicators (WGI) published by the World Bank [66]. This measure mainly indicates individuals' confidence in social rules and the degree of compliance with the rules, in particular regarding law enforcement. This indicator is perceived as a comprehensive indicator that has great influence on and is widely used in current quantitative research on governance. The indicator used for investment protection was mainly obtained from the ICRG published by the PRS Group [67]. We chose the investment profile the year before OFDI as the indicator for the level of investment protection in the host country, and the risk rating assigned was the sum of three subcomponents: contract viability/expropriation, profits repatriation, and payment delays. A lower score indicates a lower degree of investment protection, and vice versa. In terms of cultural distance, Hofstede's five dimensions, i.e., collectivism/individualism, long-term and short-term orientations, power distance, masculinity/femininity, and uncertainty avoidance, were used to measure the culture of each country. Based on the above dimensions, we could calculate the cultural distance between China and a host country [68].

$$CD_{C \cdot w} = \frac{\sum\limits_{x=1}^{5} \left[ (I_{xC} - I_{xw})^2 / V_x \right]}{5}. \tag{1}$$

In Equation (1), $CD_{C \cdot w}$ denotes the distance between China and the host country w, $I_{xC}$ denotes the score in dimension $x$, $I_{xw}$ denotes the score of the host country $w$ in dimension $x$, $V_x$ and denotes the variance in dimension $x$.



Finally, we developed a predicted $Fit_n$ variable. If the firm used the entry mode predicted by one of three theoretical models, (1) "the group response", (2) "the host-country response" and (3) "the dual response", then $Fit_n$ = 1; otherwise, the value is 0.

## 3.4. Moderating Variable

Improvement in the economic uncertainty and investment protection in host countries can have a relatively important influence on the innovation transfer of EM-MNEs' subsidiaries. Thus, to examine the moderating role of market selection and to consider the differences between the economic levels of host countries while also considering differences in the host countries calculated by determining the level of the host countries' economic development, the OECD recognizes 24 traditionally economically developed countries; if a country that received investment was one of these 24 countries, then developed country = 1; otherwise, developed country = 0.

## 3.5. Control Variables

The control variable of firm size was calculated by the natural logarithm of the firm's overall assets the year before overseas investment. Furthermore, following Shaver [62] and Brouthers et al. [31], we included a correction for the self-selection control variable using Heckman's procedure in a performance regression equation. This variable, self-selection correction (also known as "inverse Mills ratio"), was included because other studies provided evidence that unobservable firm characteristics can affect both mode choice and subsequent performance [62]. This variable was calculated based on the estimated parameters for entry mode choice in the regression model; thus, all variables in the model were used to estimate the inverse Mills ratio. The self-selection correction variable is significant because it indicates that unobservable firm characteristics are related to mode choice and subsequent performance [31]. Table 1 presents the definitions of the main variables.

**Table 1.** Definitions of the main variables.

| Variable | Definition |
| --- | --- |
| Entry mode | 1 for JV mode and 0 for WOS mode |
| Innovation performance | Natural logarithm of the number of annual patent applications for new inventions by the parent company in the three years after OFDI |
| R&D input | Ratio of R&D expenditures to the firm's total operation income in three years after OFDI |
| Asset specificity | Ratio of fixed assets to total assets of a sample firm the year before OFDI |
| Ownership | 1 for state-owned firm and 0 for private firm |
| Innovation orientation | Ratio of the number of R&D personnel to the number of all staff in the firm the year before OFDI |
| Early international experience | Ratio of overseas operational revenue to total income in the year prior to OFDI |
| Breadth | Number of foreign countries in which a firm has subsidiaries the year before OFDI |
| Depth | The number of times that the firm set up a subsidiary in the same country the year before OFDI |
| Economic uncertainty | Rolling variance of GDP growth rate over a seven-year period the year before OFDI |
| Rule of law | Indicator of "rule of law" in the WGI of the World Bank |
| Investment protection | Host country's investment profile the year before OFDI; data were obtained from the ICRG published by the PRS Group |
| Culture distance | Square root of the sum of the squared differences of the score for each cultural dimension; this value was arithmetically averaged |
| Self-selection correction | Inverse Mills ratio calculated from the estimated parameters of the entry mode choice equation, see Section 3.1 of this paper |
| Predicted $Fit_n$ ($n$ = 1, 2, or 3) | 1 if the firm used the entry mode predicted by one of three theoretical models, otherwise, the value is 0. |

JV—joint venture; WOS—wholly owned subsidiary; R&D—research and development; OFDI—outward foreign direct investment; GDP—gross domestic product; WGI— World Governance Indicator; ICRG—International County Risk Guide; PRS—Political Risk Services.

*3.6. Model Specification*

For H1 and H2, we employed Model 1 to investigate the entry mode choice:

$$\text{JVs}_{i,t} = \beta_0 + \beta_1 \text{Ownership}_{i,t-} + \beta_2 \text{Asset specificity}_{i,t-} + \beta_3 \text{Innovation orientation}_{i,t-} + \beta_4 \text{Early international experience}_{i,t-}$$
$$+ \beta_5 \text{Strategic flexibility}_{i,t-} + \beta_6 \text{Economic uncertainty}_{i,t-} + \beta_7 \text{Rule of law}_{i,t-} + \beta_8 \text{Culture distance}_{i,t-} + \mu_{i,t}. \quad (1)$$

To test H3, Model 2 was carried out to examine the relationship between internationalization and innovation performance:

$$\text{Patent}_{i,t} = \beta_0 + \beta_1 \text{Fitn}_{i,t-} + \beta_2 \text{JVs}_{i,t} + \beta_3 \text{Early international experience}_{i,t-} + \beta_4 \text{Breadth}_{i,t-} + \beta_5 \text{Depth}_{i,t-} +$$
$$\beta_6 \text{Economic uncertainty}_{i,t-} + \beta_7 \text{Investmetn protection}_{i,t-} + \beta_8 \text{Culture distance}_{i,t-} + \beta_9 \text{Self} - \text{selection correction}_{i,t} +$$
$$+ \beta_{10} \text{ R\&D input}_{i,t} + \beta_{11} \text{Ownership}_{i,t} + \beta_{12} \text{Firm scale}_{i,t-} + \beta_{13} \text{Self} - \text{selection correction}_{i,t} + \varepsilon_{i,t}. \quad (2)$$

To test H4 and H5, we used Model 3 to examine the moderating role of the host country's economic level in the relationship among economic uncertainty, investment protection, and innovation performance:

$$\text{Patent}_{i,t} = \beta_0 + \beta_1 \text{Economic uncertainty}_{i,t-} + \beta_2 \text{Investmetn protection}_{i,t-} + \beta_3 \text{Developed}_{i,t} \times \text{Economic uncertainty}_{i,t-} +$$
$$\beta_4 \text{Developed}_{i,t} \times \text{Investmetn protection}_{i,t-} + \beta_5 \text{JVs}_{i,t} + \beta_6 \text{Early international experience}_{i,t-} + \beta_7 \text{Breadth}_{i,t-} + \beta_8 \text{Depth}_{i,t-} +$$
$$+ \beta_9 \text{Culture distance}_{i,t-} + \beta_{10} \text{ R\&D input}_{i,t} + \beta_{11} \text{Ownership}_{i,t} + \beta_{12} \text{Firm scale}_{i,t-} + \beta_{13} \text{Self} - \text{selection correction}_{i,t} + \varepsilon_{i,t}, \quad (3)$$

where i and t represent the firm and year, respectively; $\beta_{0-13}$ represent the presumed parameters; and u and ε denote the measurement error terms.

## 4. Data Analysis

Hierarchical multiple logistic regression analysis was used to test our entry mode choice hypotheses. Following Brouthers et al. [31], we used a two-stage ordinary least squares (OLS) regression analysis to test our performance hypothesis. In the first step, we identified the predictors of the decision and determined whether firms made the theoretically predicted mode choice. We developed a predicted $\text{Fit}_n$ variable that takes the value of 1 if firms used the mode choice predicted by one of three theoretical models, (1) "the group response model" (which includes group response variables), (2) "the host-country response model" (which includes host-country response variables), and (3) "the dual response mode" (which includes group response and host-country response variables together), and takes the value of 0 for firms making a different mode choice.

In step two, we examined the performance implications of mode choice decisions by regressing performance on the predicted fit variable. In addition, we included a number of other variables that previous studies found to influence performance, including mode type (either JV or WOS), early international experience, international breath, international depth, economic uncertainty, investment protection, R&D input, ownership, and firm scale. We also included the correction for the self-selection variable estimated using Heckman's approach [62].

In step three, we examined the moderating effect of market selection on firms' innovation performance; using the interaction item of the economic uncertainty of the host country and entering a developed economy, and the interaction item of the investment guarantee level of the host country and entering a developed country, we tested the moderating effect of entering a developed country on the relationships among the economic uncertainty of the host country, and the level of investment protection and innovation performance.

The correction for the self-selection variable is important because unobservable firm characteristics may be related to mode choice and performance; e.g., firms may choose strategies based on unmeasured attributes and industry conditions, making strategic choice endogenous and self-selected. Thus, we followed Heckman's advice and included the correction for the self-selection variable. By controlling for self-selection using Heckman's approach, we controlled for the possible misspecification of the performance models [62].

## 5. Results

### 5.1. Descriptive Statistics and Correlation Analysis

Table 2 shows the descriptive statistics and correlation matrix of the main variables. A correlation analysis was conducted before the regression was run. We also checked the variance inflation factors (VIFs) and found that they were all less than 10, indicating that multicollinearity was not a major issue in our study.

### 5.2. Analysis of Firms' Entry Mode Choice

The results are shown in Table 3. Model 1 (the group response mode) in Table 3 examined only the variable for group response, which was statistically significant ($p < 0.01$). Model 2 (the host-country response mode) in Table 3 examined only the variable for host-country response, which was statistically significant ($p < 0.01$). Model 3 (the dual response mode) in Table 3 shows the model that integrated the factors of the group response and the host-country response, which was statistically significant ($p < 0.01$).

Next, the contrast test of the likelihood between nested models lrtest:-2[L(red)–L(full)] was conducted, and the results showed that the chi-square value ($\chi^2$) of Model 3 to Model 1 was 13.82 ($p < 0.01$). The chi-square value ($\chi^2$) of Model 3 to Model 2 was 16.93 ($p < 0.01$). In other words, Model 3 has stronger explanatory power than Model 1 or Model 2.

Furthermore, it can be inferred from Model 3 (which has good explanatory power) that state ownership is positively related to the JV mode ($p < 0.01$); asset specificity is positively related to the WOS mode ($p < 0.10$); early international experience is positively related to the JV mode ($p < 0.10$); economic uncertainty has a more positive influence on the JV mode ($p < 0.10$); and better rule of law is positively related to the WOS mode ($p < 0.05$). The main parts of H1 and H2 were supported by the results. The results are consistent with the research conclusions of Brouthers et al. [2,4], Lu et al. [47], and Mingo et al. [49].

In addition, it can be seen that Model 3 correctly predicted 72.86% of the mode choices, which was substantially more than that of either Model 1 or Model 2. This result means that the entry mode driven by dual response better predicts Chinese firms' entry mode choices. The results are similar to the research results of Brouthers et al. [31].

**Table 2.** Descriptive statistics and correlation analysis.

| Variables | Mean | SD | 1 | 2 | 3 | 4 | 5 | 6 | 7 | 8 | 9 | 10 | 11 | 12 | 13 | 14 | 15 |
|---|---|---|---|---|---|---|---|---|---|---|---|---|---|---|---|---|---|
| Entry mode | 0.26 | 0.44 | 1 | | | | | | | | | | | | | | |
| Innovation Performance | 3.91 | 1.87 | −0.121 * | 1 | | | | | | | | | | | | | |
| R&D input | 0.04 | 0.03 | −0.006 | 0.143 ** | 1 | | | | | | | | | | | | |
| Ownership | 0.39 | 0.49 | 0.187 *** | 0.129 ** | −0.056 | 1 | | | | | | | | | | | |
| Asset specificity | 0.23 | 0.13 | −0.093 | −0.216 *** | −0.298 *** | −0.067 | 1 | | | | | | | | | | |
| Innovation orientation | 0.19 | 0.12 | −0.037 | 0.012 | 0.411 *** | 0.144 ** | −0.248 *** | 1 | | | | | | | | | |
| Early international experience | 0.27 | 0.29 | 0.048 | −0.155 ** | −0.073 | −0.353 *** | 0.063 | −0.154 ** | 1 | | | | | | | | |
| Breadth | 1.48 | 2.35 | 0.037 | 0.331 *** | 0.054 | 0.106 * | −0.147 ** | 0.015 | −0.037 | 1 | | | | | | | |
| Depth | 0.94 | 1.58 | 0.051 | 0.200 *** | 0.178 *** | 0.04 | −0.255 *** | 0.036 | −0.062 | 0.586 *** | 1 | | | | | | |
| Economic uncertainty | 0.07 | 0.05 | 0.198 *** | 0.022 | 0.058 | 0.026 | −0.081 | 0.019 | −0.032 | 0.222 *** | 0.152 ** | 1 | | | | | |
| Rule of law | 82.13 | 18.87 | −0.206 *** | −0.088 | 0.065 | −0.046 | −0.043 | 0.086 | −0.024 | −0.161 ** | 0.033 | −0.420 *** | 1 | | | | |
| Investment protection | 10.92 | 1.53 | −0.191 *** | −0.012 | −0.045 | 0.006 | −0.038 | 0.057 | −0.026 | −0.158 ** | −0.025 | −0.468 *** | 0.774 *** | 1 | | | |
| Culture distance | 3.2 | 1.89 | 0.021 | −0.029 | 0.128 ** | −0.047 | 0.014 | 0.068 | 0.063 | 0.138 ** | 0.138 ** | 0.251 *** | 0.07 | −0.025 | 1 | | |
| Firm size | 21.89 | 1.14 | 0.063 | 0.438 *** | −0.183 *** | 0.410 *** | −0.111 * | −0.06 | −0.292 *** | 0.447 *** | 0.310 *** | 0.189 *** | −0.128 ** | −0.099 | 0.01 | 1 | |
| Self-selection correction | 1.34 | 0.38 | −0.341 *** | −0.088 | 0.043 | −0.568 *** | 0.333 *** | 0.078 | −0.147 ** | −0.06 | −0.129 ** | −0.512 *** | 0.536 *** | 0.444 *** | −0.062 | −0.261 *** | 1 |
| Fit$_2$ | 0.74 | 0.44 | −0.893 *** | 0.028 | −0.005 | −0.151 ** | 0.097 | 0.068 | −0.062 | −0.105 | −0.052 | −0.235 *** | 0.245 *** | 0.166 *** | −0.045 | −0.061 | 0.337 *** |

Notes: * $p < 0.10$, ** $p < 0.05$, *** $p < 0.01$.

**Table 3.** Logistic regressions for entry mode. LR—likelihood ratio.

| Variables | Model 1 | Model 2 | Model 3 |
|---|---|---|---|
| *Group response variables* | | | |
| Ownership | 1.175 *** | | 1.238 *** |
| | (0.344) | | (0.359) |
| Asset specificity | −2.110 * | | −2.233 * |
| | (1.244) | | (1.296) |
| Early international experience | 1.152 ** | | 1.242 ** |
| | (0.584) | | (0.609) |
| Strategic flexibility | −0.056 | | −0.264 |
| | (0.199) | | (0.217) |
| Innovation orientation | −1.488 | | −1.170 |
| | (1.309) | | (1.333) |
| *Host-country response variables* | | | |
| Economic uncertainty | | 5.567 * | 6.045 * |
| | | (3.174) | (3.278) |
| Rule of law | | −0.016 ** | −0.019 ** |
| | | (0.008) | (0.009) |
| Culture distance | | −0.005 | 0.024 |
| | | (0.087) | (0.092) |
| Constant | −1.085 * | −0.164 | 0.039 |
| | (0.566) | (0.778) | (1.037) |
| Number of observations | 242 | 242 | 242 |
| Chi$^2$ | 15.91 | 12.80 | 29.73 |
| Pseudo $R^2$ | 0.058 | 0.047 | 0.108 |
| Likelihood | −129.8 | −131.3 | −122.8 |
| LR test based on Model 1 | | | 13.82 *** |
| LR test based on Model 2 | | | 16.93 *** |
| Overall prediction accuracy | 65.87% | 64.39% | 72.86% |

Notes: * $p < 0.10$, ** $p < 0.05$, *** $p < 0.01$; *t*-values are in parentheses.

*5.3. Entry Mode Choice and Innovation Performance*

Table 4 shows the results of the OLS regressions that use parent innovation performance as a dependent variable. The VIFs relating to multicollinearity in each model were all below the tolerance level of 10, indicating that the relevant models have relatively good explanatory power. We conducted four analyses on the regression model for parent innovation performance: Model 1 did not include the predicted fit variable (Fit) but included internationalization and other control variables; Model 2 included the predicted Fit$_1$ variable for group response, internationalization, and the other control variables; Model 3 included the predicted Fit$_2$ variable of host-country response internationalization and other control variables; and Model 4 included the predicted Fit$_3$ variable, which integrates both the group response and the host-country response variables, internationalization, and the other control variables. All four performance models were highly significant ($p < 0.01$).

**Table 4.** Ordinary least squares (OLS) regressions for innovation performance.

| Variables | Model 1 | Model 2 | Model 3 | Model 4 |
|---|---|---|---|---|
| $Fit_1$ | | 0.381 (0.727) | | |
| $Fit_2$ | | | −1.171 ** (0.524) | |
| $Fit_3$ | | | | −0.371 (0.353) |
| JV | −0.662 *** (0.244) | −0.308 (0.719) | −1.704 *** (0.525) | −0.924 *** (0.349) |
| Early international experience | −1.339 ** (0.517) | −1.341 ** (0.518) | −1.228 ** (0.515) | −1.270 ** (0.521) |
| Breadth | 0.252 *** (0.063) | 0.252 *** (0.064) | 0.216 *** (0.065) | 0.242 *** (0.064) |
| Depth | −0.197 ** (0.087) | −0.196 ** (0.087) | −0.165 * (0.087) | −0.187 ** (0.087) |
| Economic uncertainty | −9.156 *** (3.009) | −9.024 *** (3.025) | −9.174 *** (2.984) | −8.590 *** (3.056) |
| Investment protection | 0.183 ** (0.087) | 0.179 ** (0.087) | 0.148 * (0.087) | 0.177 ** (0.087) |
| Culture distance | −0.050 (0.056) | −0.051 (0.056) | −0.051 (0.056) | −0.050 (0.056) |
| R&D input | 15.439 *** (3.603) | 15.473 *** (3.610) | 15.210 *** (3.574) | 15.220 *** (3.609) |
| Ownership | −1.429 *** (0.411) | −1.429 *** (0.412) | −1.284 *** (0.413) | −1.372 *** (0.415) |
| Firm scale | 0.711 *** (0.116) | 0.710 *** (0.116) | 0.725 *** (0.115) | 0.711 *** (0.116) |
| Self-selection correction | −2.359 *** (0.620) | −2.343 *** (0.621) | −2.116 *** (0.624) | −2.229 *** (0.632) |
| Constant | −9.349 *** (2.841) | −9.687 *** (2.918) | −8.530 *** (2.841) | −9.185 *** (2.845) |
| Number of observations | 242 | 242 | 242 | 242 |
| F | 10.55 | 9.665 | 10.26 | 9.770 |
| $R^2$ | 0.335 | 0.336 | 0.350 | 0.339 |
| Adjusted $R^2$ | 0.304 | 0.301 | 0.316 | 0.304 |

Notes: * $p < 0.10$, ** $p < 0.05$, *** $p < 0.01$; *t*-values are in parentheses.

In terms of the goodness of fit, $Fit_1$ and $Fit_3$ did not show that there were close relationships between the innovation performance in Model 2 and Model 4, while $Fit_2$ was negatively related ($p < 0.05$) to the innovation performance in Model 3. Therefore, H3 did not pass the test. In addition, the entry mode that responds to the host country unilaterally ($Fit_2$) had a significantly negative influence on innovation performance. This is possible because, under the influence of economic crises in developed countries, the return of Western manufacturing, and the anti-globalization movement, EM-MNEs' entry mode choices are greatly influenced by the host-country environment. For example, for a JV model with a more general host-country response tendency, if Chinese firms lack the capabilities of cross-cultural management and innovative network integration required by this mode, their innovation performance would be adversely impacted [69].

Finally, in the leapfrog stage of internationalization development, the choice of entry mode driven by dual response ($Fit_3$) fails to significantly correlate with innovation performance. This result occurs because Chinese firms generally lack a dynamic internationalization capability required by such entry

modes. In other words, these firms lack the ability to adapt to changes in the internal and external environment, institute flexible changes in their existing behavior patterns and organizational practices, and choose the optimal organizational form and governance mechanism among various entry modes. Although the entry mode choice represents the best entry mode choice for Chinese firms to use in the future, this mode does not reflect the current actual "profit model" and firms' internationalization dynamic capabilities need further improvement [70].

The consistent conclusions shown in Table 4 indicate that early international experience was negatively related ($p < 0.05$) to parent innovation performance from Model 1 to Model 4. This result occurs because most Chinese firms' early international experience is mainly influenced by the two macro policies of "foreign direct investment" and "export orientation" [71]. As a result, Chinese firms based on "export trade" accumulate a "low level" of international experience in the early internationalization stage; Chinese firms have insufficient high-level internationalization capabilities in the OFDI phase, for example, the ability to balance market development and technological innovation. Thus, the accumulation of such a low level of early international experience will weaken their innovation performance. The breadth of international experience is positively related ($p < 0.01$) to innovation performance in all four models, potentially due to the integration capability of innovation resources related to multi-regional business investment (external socialization). In other words, structural absorption (cooperation networks) and external absorption (culture and values) have a significant influence on the innovation behavior of subsidiaries and parents. The depth of international experience is negatively related ($p < 0.05$) to innovation performance in Models 1, 2, and 4, and negatively related ($p < 0.10$) to innovation performance in Model 3. This result proves that, during the present period when Chinese firms are experiencing the change from applied innovation to explorative innovation, innovation and market resources are repeatedly invested in specific areas that pose obstacles for Chinese firms in terms of their ability to fully integrate multi-level innovation resources. Additionally, economic uncertainty is negatively related ($p < 0.01$) to innovation performance in all four models. Investment protection is positively related ($p < 0.05$) to innovation performance in Models 1, 2, and 4, and positively related ($p < 0.10$) to innovation performance in Model 3, which supports H4. The results are consistent with the research conclusions of Wu et al. [52].

We also found that R&D input is positively related ($p < 0.01$) to innovation performance in all four models, meaning that the R&D input of firms that Chinese manufacturers invest in has positive effects on technological innovation and intellectual property. State ownership is negatively related ($p < 0.01$) to innovation performance in all four models, meaning that, compared with state-owned firms that have more politically related goals, private firms have more freedom in terms of their innovation strategy and have better innovation performance. Firm size is positively related ($p < 0.01$) to innovation performance in all four models, meaning that the scale economy of innovation in Chinese firms is relatively significant. The correction for the self-selection variable is negative and significantly ($p < 0.01$) related to firm' innovation performance in all four models, indicating that unobservable firm characteristics are related to mode choice and performance.

*5.4. Market Slection and Inovation Performance: The Moderating Role of the Host Country's Economic Level*

H4 predicts that a stable economic environment and good investment protection in the host country enable EM-MNEs to have better innovation performance. As shown in the results from Model 2 to Model 6 in Table 5, the coefficient of economic uncertainty is negative and highly significant ($p < 0.01$), and the coefficient of investment protection is positive and significant (in Model 2 and Model 3, $p < 0.05$; in Model 4 and Model 6, $p < 0.01$). Thus, the result again supports H4. H5 predicts that compared with a stable economic foundation and balanced investment protection in developed countries, the socioeconomic conditions of developing countries play a more significantly sensitive role in moderating economic uncertainty, investment protection, and innovation performance. As shown in the analysis results from Model 4 to Model 6 in Table 4, the coefficient of the interaction term between economic uncertainty and developed country (Model 4) is positive and highly significant ($p < 0.1$),

and the coefficient of the interaction term between investment protection and developed country (Model 5) is negative and significant ($p < 0.05$). The results for the two moderating effects remain consistent in Model 6, which includes all variables and interaction terms.

**Table 5.** International market selection and innovation performance.

| Variables | Model 1 | Model 2 | Model 3 | Model 4 | Model 5 | Model 6 |
|---|---|---|---|---|---|---|
| Economic uncertainty | | −9.156 *** | −8.923 *** | −15.310 *** | −9.243 *** | −16.082 *** |
| | | (3.204) | (3.303) | (4.076) | (3.271) | (3.754) |
| Investment protection | | 0.183 ** | 0.214 ** | | 0.586 *** | 0.589 *** |
| | | (0.089) | (0.095) | | (0.197) | (0.187) |
| Developed country | | | −0.246 | −0.765 | 4.130 * | 2.090 |
| | | | (0.494) | (0.686) | (2.212) | (2.238) |
| Economic uncertainty × developed country | | | | 9.836 * | | 11.779 ** |
| | | | | (5.061) | | (4.702) |
| Investment protection × developed country | | | | | −0.482 ** | −0.408 * |
| | | | | | (0.226) | (0.214) |
| Culture distance | −0.082 | −0.050 | −0.043 | −0.039 | −0.062 | −0.043 |
| | (0.054) | (0.054) | (0.056) | (0.056) | (0.056) | (0.055) |
| JV | −0.701 *** | −0.662 *** | −0.666 *** | −0.656 *** | −0.671 *** | −0.645 *** |
| | (0.252) | (0.244) | (0.243) | (0.239) | (0.243) | (0.238) |
| Early international experience | −0.344 | −1.339 *** | −1.243 ** | −1.189 ** | −1.208 ** | −1.251 ** |
| | (0.434) | (0.456) | (0.535) | (0.541) | (0.528) | (0.520) |
| Breadth | 0.155 *** | 0.252 *** | 0.242 *** | 0.224 *** | 0.235 *** | 0.234 *** |
| | (0.060) | (0.064) | (0.074) | (0.074) | (0.073) | (0.072) |
| Depth | −0.097 | −0.197 *** | −0.184 ** | −0.186 ** | −0.178 ** | −0.201 ** |
| | (0.077) | (0.074) | (0.083) | (0.086) | (0.082) | (0.083) |
| R&D input | 14.405 *** | 15.439 *** | 15.479 *** | 14.760 *** | 15.444 *** | 15.368 *** |
| | (2.807) | (2.625) | (2.619) | (2.720) | (2.655) | (2.595) |
| Ownership | −0.412 | −1.429 *** | −1.352 *** | −1.174 ** | −1.311 *** | −1.314 *** |
| | (0.317) | (0.387) | (0.466) | (0.473) | (0.459) | (0.457) |
| Firm scale | 0.702 *** | 0.711 *** | 0.713 *** | 0.726 *** | 0.707 *** | 0.724 *** |
| | (0.125) | (0.121) | (0.122) | (0.119) | (0.122) | (0.119) |
| Self-selection correction | −0.565 | −2.359 *** | −2.214 *** | −1.821 ** | −2.156 *** | −2.029 *** |
| | (0.409) | (0.623) | (0.779) | (0.802) | (0.756) | (0.757) |
| Constant | −10.686 *** | −9.349 *** | −9.827 *** | −7.950 *** | −12.823 *** | −12.568 *** |
| | (2.845) | (2.814) | (3.156) | (2.934) | (3.248) | (3.034) |
| Number of observations | 242 | 242 | 242 | 242 | 242 | 242 |
| F | 12.96 | 12.62 | 12.41 | 10.61 | 11.35 | 11.47 |
| $R^2$ | 0.296 | 0.335 | 0.336 | 0.337 | 0.347 | 0.364 |
| Adjusted $R^2$ | 0.268 | 0.304 | 0.302 | 0.302 | 0.310 | 0.325 |

Notes: * $p < 0.10$, ** $p < 0.05$, *** $p < 0.01$; $t$-values are in parentheses.

To gain further insight into these findings, we plotted the related interaction results based on Model 4 (Model 4 in Table 5). Figure 1 illustrates the interaction between market selection and economic uncertainty in the host country. As shown in Figure 1, the slope is more negative when a firm invests in a developing country. In contrast, the slope becomes less negative when a firm invests in a developed country. This contrast clearly shows that the relationship between economic uncertainty and innovation performance becomes more negative when a firm invests in a developing country, which provides strong support for H5.

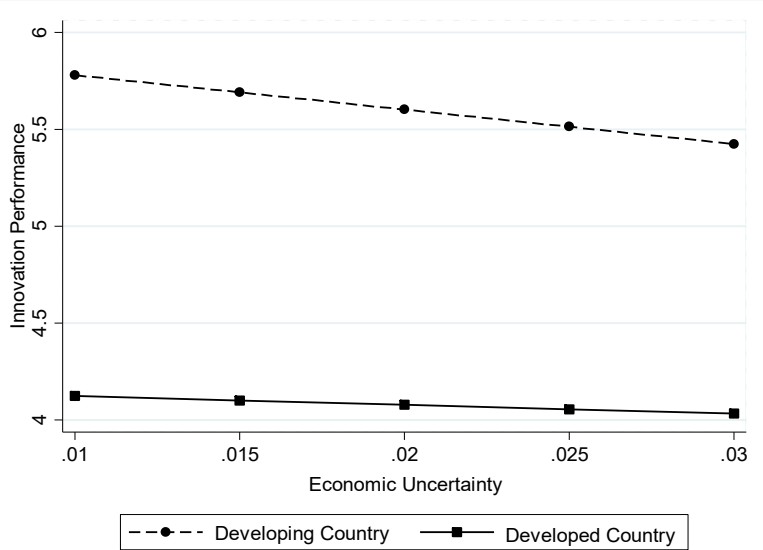

**Figure 1.** The moderating role of market selection on the relationship between economic uncertainty and innovation performance.

To gain further insight into these findings, we plotted the related interaction results based on Model 5 (Model 5 in Table 5). Figure 2 illustrates the interaction between market selection and investment protection in the host country. As shown in Figure 2, the slope is more positive when a firm invests in a developing country. In contrast, the slope becomes less positive when a firm invests in a developed country. This contrast clearly shows that the relationship between investment protection and parent innovation performance becomes more positive when a firm invests in a developing country, which provides strong support for H5.

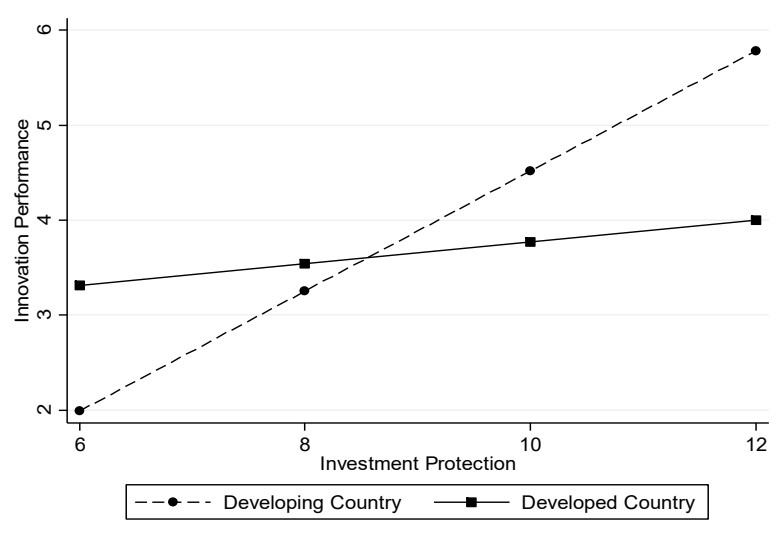

**Figure 2.** The moderating role of market selection on the relationship between investment protection and innovation performance.

## 6. Discussion

While the relationship between EM-MNEs' internationalization and their performance is an emerging topic, it still requires additional investigation. We firstly integrated the research on EM-MNEs' entry mode choice and innovation performance in the same theoretical framework and empirical test, and found that some unobserved organizational characteristics had endogenous test problems caused

by the self-selection of the samples. Our empirical findings showed that the relationship between EM-MNEs' entry mode choice and innovation performance was different from Western scholars' research results. The entry mode driven by the dual response to group and host-country consistency could not significantly enhance EM-MNEs' innovation performance, but the entry mode that is unilaterally driven by the host country response has a significant negative influence on EM-MNEs' innovation performance. When we further divided the indicators of international experience into early international experience, international breadth, and international depth, we found that different aspects of internationalization had different influences on EM-MNEs' innovation performance, which could explain the conflict in the conclusions of relevant studies. Finally, we found that different socioeconomic conditions in the regions where EM-MNEs invest have different moderating effects on the host-country institutions and innovation performance. Our research enriches relevant studies on the relationship between internationalization and EM-MNEs' performance and provides inspiring empirical evidence.

There are some limitations of our research. Firstly, because the internationalization of Chinese manufacturers is still at an early stage, the size and quality of the samples are not perfect. The number of listed manufacturing firms in Shanghai and Shenzhen that can be surveyed and chosen is limited. Moreover, non-listed firms in emerging markets have many problems (for example, poor corporate governance and information disclosure); thus, we could not select them as samples in the empirical test to guarantee the test validity. Nevertheless, with an improvement in management and information disclosure in these firms, we will expand the range of samples to non-listed manufacturing firms to expand the sample size and the applicability of the results. Secondly, in future directions of the research, considering that Chinese manufacturers emphasize growth and innovation in the early period of internationalization (i.e., foreign subsidiaries of EM-MNEs have the mission of entering a global innovation network and transferring knowledge and technology), we will further focus on the relationship between internationalization and parent innovation performance. However, considering EM-MNEs' dual strategy in terms of market growth and technology innovation, research on the relationship between EM-MNEs' internationalization and their performance will further divide performance into innovation performance and financial performance in the same period. Only through this distinction can we accurately grasp the characteristics of the dual mission and then reveal how EM-MNEs' internationalization factors "jointly promote" or "offset each other" in the two strategic performances.

## 7. Conclusions and Implications

Combining theories of the RC perspective and IB perspective, this paper designed three kinds of theoretical predicting models for EM-MNEs' entry modes: the predicting model of group response, the predicting model of host-country response, and the predicting model of the dual response to both group and host-country consistency. We also assessed the impact of EM-MNEs' internationalization and host-country institutional factors on innovation performance. We further examined the moderating role of the economic level of the invested regions on the relationships among economic uncertainty, investment protection, and parent innovation performance. Based on the samples of OFDI projects of Chinese manufacturers from 2003 to 2014, we found the following:

(1) State-owned firms tend to choose JV modes that have high market entry efficiency, while private firms tend to choose WOS entry modes with intra-group consistency. Asset specificity enhances internal consistency, which has a positive influence on the WOS entry mode. When they accumulate early international experience, firms tend to choose the JV mode with more openness. Economic uncertainty and low investment protection of the host country have a significant positive influence on the choice of the JV mode. The entry mode that integrates group response and host-country response has better results in the prediction of entry modes of Chinese firms. These conclusions are in line with the Western research results based on the IB perspective.

(2)　The results of this study on the relationship between the entry mode choice of Chinese firms and performance differ from the results in Western theoretical research. Currently, the entry mode response to the host country unilaterally has a significantly negative influence on EM-MNE performance. That is, when the international environment is complicated, the pressure of host-country legitimacy exerts increasing influence on entry mode choices. As a result, EM-MNEs are unilaterally subject to external environment pressure and choose the entry mode responding to host-country consistency. However, the adjustment of EM-MNEs' structure and organizational capability generally lags behind the need for international innovation, and the ability to successfully engage in cross-cultural management and innovation network integration cannot be improved in the short term. Correspondingly, the high-level internationalization capability of balancing market development and innovation exploration is also lacking, which negatively affects innovation performance.

(3)　The results of this study regarding the effect of a group's internal factors and the host country's institutional factors on innovation performance is consistent with results found by Western scholars. R&D input, firm size, the breadth of international experience, economic stability of the host country, and investment protection are positively related to innovation performance; self-selection correction can also enhance the predictive ability of the regression models. However, there are areas of this research that differ from Western research. For example, state ownership, early international experience based on foreign market expansion, and the depth of international experience based on repeated investment in the same region have a significant negative influence on innovation performance.

(4)　We also examined and confirmed the moderating effect of different socioeconomic conditions in invested regions on the relationships among economic fluctuation, investment protection, and parent innovation performance. The results enrich studies on the relationship between internationalization and the performance of EM-MNEs, and provide some guidance for EM-MNEs' practices.

The results of this paper can give some advice for EM-MNEs' strategic choices and innovation practices. In practice, the theoretical entry mode driven by the dual response to group and host-country consistency has better predictive ability, and it also represents a future trend of the strategic choices of EM-MNEs. However, EM-MNEs' behavior that conforms to the best model predictions improves innovation outcomes, but depends on the growth of their international capability. Therefore, EM-MNEs also need to enhance their dynamic capabilities to match their entry mode choice and need to try avoid the entry mode that is unilaterally driven by the host-country response because it may impair their innovation performance. When increasing R&D input, EM-MNEs can consider the good socioeconomic and institutional conditions in developed countries to enhance their innovation performance. Specifically, EM-MNEs can better develop the integration capability of multi-regional business to improve the development of high-quality, multi-level innovation resources in regions that have abundant resources for innovation. Finally, EM-MNEs need to avoid the negative influence of state ownership, the low level of early international experience, and excessive investment in a single region on their innovation performance. Meanwhile, EM-MNEs should focus on the moderating effects of the different socioeconomic conditions in the markets where they invest on the relationship between the institutional factors of the host country and their performance.

**Author Contributions:** C.W. designed the research, analyzed the data and drafted the manuscript; F.H. put forward the valuable suggestions, analyzed the data and reviewed and edited the draft; C.H. provided the data and analyzed the data; H.Z. put forward the valuable suggestions.

**Funding:** This research was funded by the Jiangsu Planning Office of Philosophy and Social Science (18GLB025), the National Social Science Fund of China (17BGL200), and China Postdoctoral Science Foundation (2014M561608).

**Conflicts of Interest:** The authors declare no conflict of interest.

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
