# Peer review of "Entry Mode, Market Selection, and Innovation Performance"

_sustainability, doi:10.3390/su10114222_

Reviewer 1 Report

The paper has sound mathematical approach to very interesting topic. 

However, more accent should be put on the novelty of the research and up-to-date references. 

Data analysis depicts several partial explanations of the analysis, but is lacking the depiction of the methodology of the research. 

Logistic regression does not follow many of the key assumptions of linear regression and general linear models, but  some assumptions are applied. The problem of multicollinearity was solved in the study, however, logistic regression requires a large sample size - does the sample size meet the general guideline of minimal cases per independent variable of the model? 

Results are explained properly. Please, provide data on similar researches and their results. 

Discussion - please, include all limitations of the research, future directions of the research, comparison with results of other similar / compatible studies. 

Conclusion - please, state what this research and its results can give to the economy / government/ entrepreneurs. 

Comments: 

     line 96, 97 - repetition of "this paper"

     line 100 - "firms´s" instead of "firms´ "

     line 354, 355 - please explain clearly the research period 

Author Response

Response to Reviewer 1 Comments

Point 1: However, more accent should be put on the novelty of the research and up-to-date references. 

Response 1: We have added some representative literature in recent years and even in 2018.

Point 2: Data analysis depicts several partial explanations of the analysis, but is lacking the depiction of the methodology of the research. 

Response 2: We have described the research method of this study step by step in "4. Data Analysis" and described how to use these methods.

Point 3: Logistic regression does not follow many of the key assumptions of linear regression and general linear models, but some assumptions are applied. The problem of multicollinearity was solved in the study, however, logistic regression requires a large sample size - does the sample size meet the general guideline of minimal cases per independent variable of the model? 

Response 3: In this case, the entry mode is defined as a binary variable, so, we use the logistic model as the parameter estimation method. Compared with OLS estimation, logistic regression independent variables do not need to satisfy the hypothesis of homoscedasticity and multivariate normal distribution.

In theory, logistic regression adopts the maximum likelihood estimation, which has many advantages. However, one major disadvantage is that there must be enough samples to ensure its validity. Therefore, we screened 8 key independent variables as little as possible to ensure that the regression sample size is 15-20 times the number of independent variables required (see "applied logistic regression/David W. Hosmer, Jr., Stanley Lemeshow.─2nd ed.", P346). At the same time, we performed the regression test with Probit and OLS, and the results were all valid and basically consistent.

Point 4: Results are explained properly. Please, provide data on similar researches and their results. 

Response 4: In the revised paper, we analyzed the results of each test, and we provided similar studies and conclusions from both domestic and international studies.

Point 5: Discussion - please, include all limitations of the research, future directions of the research, comparison with results of other similar / compatible studies. 

Response 5: We have added what you requested in “5. Discussion” after revised the paper.

Point 6: Conclusion - please, state what this research and its results can give to the economy / government/ entrepreneurs. 

Response 6: We have added what you requested in “6. Conclusion and Implication” after revised the paper.

Point 7: line 96, 97 - repetition of "this paper”; line 100 - "firms´s" instead of "firms´ "; line 354, 355 - please explain clearly the research period  

Response 7: We have proofread the whole paper by the professionals.

Reviewer 2 Report

From my point of view the paper objective is not clear, probably because the authors want to analyse too much aspects concerning different issues. At the same time, also the inferential analysis are too much and this prevents the reader from having a unified vision of the observed aspects. 

The hypotheses are not always supported by a consistent literature review. In the section 2 the authors often describe important subjects and theories using cryptic language and quick mentions.

A proofreading of all text is absolutely necessary, there are many typos and errors that make the work unclear.

I would suggest to the authors to focus the study on some more specific aspects, perhaps supporting them with a more coherent and deepen analysis of the literature. Inferential analysis have also be more focused. Otherwise the reader does not perceive the usefulness of the work.

Author Response

Response to Reviewer 2 Comments 

 Point 1: From my point of view the paper objective is not clear, probably because the authors want to analyse too much aspects concerning different issues. At the same time, also the inferential analysis are too much and this prevents the reader from having a unified vision of the observed aspects. 

Response 1: As experts say, we have researched more content. Namely, the impacts of entry mode and market selectionthe two types of standard strategic choices used in OFDI on innovation performance are studied at the same time.

This is because we want to better reveal the relationship between the strategic choices and innovation performance of MNEs by incorporating entry mode and market selection(enter developed countries or enter developing countries) into an integrated analytical framework[refer to article 8,9].

Namely, Once MNEs enter into the overseas market, it is not only the entry mode choice that directly influence the innovation performance, but also the relationship between the host-country institutional factors and the innovation performance is moderated by the market selection, namely the socioeconomic condition in invested area moderates the relationship between the host-country institutional factors and the MNEs  innovation performance [refer to article 9].

Point 2:  The hypotheses are not always supported by a consistent literature review. In the section 2 the authors often describe important subjects and theories using cryptic language and quick mentions. 

Response 2: In the section 2, We've added a representative literature that is consistent with the hypotheses, and, by using the representational results of this field, we have made a concrete and clarifying adjustment to the language.

Point 3: A proofreading of all text is absolutely necessary, there are many typos and errors that make the work unclear. 

Response 3: We polished many language, typos and errors after revised the paper.

Point 4: I would suggest to the authors to focus the study on some more specific aspects, perhaps supporting them with a more coherent and deepen analysis of the literature. Inferential analysis have also be more focused. Otherwise the reader does not perceive the usefulness of the work.

Response 4: In accordance with the expert advice, we added the research significance of the relationship between OFDI’s two standard strategic choices (entry mode/market selection) and innovation performance in the introduction, and deleted some content that is not closely related to this subject (such as the influence of some control variables) in the demonstration process. 

Round  2

Reviewer 2 Report

I think that the article has been improved